# Visualization Analysis of Small Extracellular Vesicles in the Application of Bone-Related Diseases

**DOI:** 10.3390/cells13110904

**Published:** 2024-05-24

**Authors:** Xinjiani Chen, Ning Yang, Bailei Li, Xinyu Gao, Yayu Wang, Qin Wang, Xiaojun Liu, Zhen Zhang, Rongqing Zhang

**Affiliations:** 1Department of Biotechnology and Biomedicine, Yangtze Delta Region Institute of Tsinghua University, Jiaxing 314006, China; 1601111432@pku.edu.cn (X.C.); anany0801@163.com (N.Y.); libl1213@foxmail.com (B.L.); 13287797632@163.com (X.G.); wangyayu521@126.com (Y.W.); lavender_412@163.com (Q.W.); txliuxj@vip.163.com (X.L.); 2Ministry of Education Key Laboratory of Protein Sciences, School of Life Sciences, Tsinghua University, Beijing 100084, China; 3Zhejiang Provincial Key Laboratory of Applied Enzymology, Yangtze Delta Region Institute of Tsinghua University, 705 Yatai Road, Jiaxing 314006, China; 4Taizhou Innovation Center, Yangtze Delta Region Institute of Tsinghua University, Jiaxing 318000, China

**Keywords:** extracellular vesicles, bone-related diseases, MSCs, tissue engineering, osteoarthritis, senescence

## Abstract

Small extracellular vesicles were shown to have similar functional roles to their parent cells without the defect of potential tumorigenicity, which made them a great candidate for regenerative medicine. The last twenty years have witnessed the rapid development of research on small extracellular vesicles. In this paper, we employed a scientometric synthesis method to conduct a retrospective analysis of small extracellular vesicles in the field of bone-related diseases. The overall background analysis consisted the visualization of the countries, institutions, journals, and authors involved in research. The current status of the research direction and future trends were presented through the analysis of references and keywords, which showed that engineering strategies, mesenchymal stem cell derived exosomes, and cartilage damage were the most concerning topics, and scaffold, osteoarthritis, platelet-rich plasma, and senescence were the future trends. We also discussed the current problems and challenges in practical applications, including the in-sight mechanisms, the building of relevant animal models, and the problems in clinical trials. By using CiteSpace, VOSviewer, and Bibliometrix, the presented data avoided subjective selectivity and tendency well, which made the conclusion more reliable and comprehensive. We hope that the findings can provide new perspectives for researchers to understand the evolution of this field over time and to search for novel research directions.

## 1. Introduction

Exosomes, or vaguely called “extracellular vesicles (EVs)”, were first discovered in the 1980s [1] and named by Johnstone et al. [2] because of their characterization of release. For a long time, the prevailing belief was that exosomes acted as garbage processors in the cell system; the inclusions were useless and were thrown out to maintain the healthy environment within cells [3]. It was not until 1996 that G. Raposo et al. [4] found that exosomes contained functional MHC (major histocompatibility complex)-class II and had a strong relationship with antigen presentation. Since then, exosomes have been found to show significant importance in the fields of cell proliferation, tissue repair, tumor diagnosis, and so on, and have caught the attention of more and more scientists, causing them to dig deeper into this promising subject.

Exosomes were generally defined as one type of extracellular vesicles with diameters ranging from 30 to 150 nm and containing abundant nucleic acid, proteins, cytokines, lipid molecules, and other compounds, and almost all cells in the body could secrete them. The other three types were microvesicles, apoptotic bodies, and oncosomes [5]. However, the isolation and purification of exosomes was not an easy process and restricted the development of their applications to some extent [6]. The ISEV (International Society for Extracellular Vesicles) board members separately announced guidelines in 2014 [7], 2018 [8], and 2023 [9] to standardize EV isolation and characterization procedures; in particular, the 2023 guidelines summarized different EV separation and characterization methods for specific uses and pointed out that tetraspanins, such as CD63, CD81, and TSG101, could not represent exosomes. Considering the fact that it was not yet possible to extract purified exosomes through contemporary isolation techniques, it was more appropriate to use small EVs (sEVs) to mimic this subtype. Thus, in this article, “small extracellular vesicles (sEVs)” were used instead of exosomes, so they could be distinguished by size (a diameter less than 200 nm) without the classification of biogenesis pathways.

With the aging of the population, the incidence of bone-related diseases has also increased in recent years, which has become a major problem threatening national health. Globally, more than one billion people experienced musculoskeletal disease in 2019, and it was also the largest contributor to disability life years worldwide [10]. It was shocking to find that this kind of disease accounted for the largest demand for rehabilitation services among children. Common bone-related diseases included joint disease (osteoarthritis, rheumatoid arthritis, etc.), skeletal disease (osteoporosis, fracture, etc.), spinal problems (spinal cord injury, cervical spondylosis, etc.), other systemic diseases (gout, ischemic necrosis, etc.), and so on. The pathogenic factors and types varied from each other and most of them could not be reversed once formed; thus, there were great difficulties in bone injury repair. In recent years, cell-based therapeutics have emerged as a promising therapeutic modality, and it was found that sEVs had the ability to penetrate the blood–brain barrier with low immunogenicity naturally, which made them a unique and valuable strategy to replace the conventional route of treatment and realize more effective targeted therapies.

The past twenty years have witnessed research bursts in the study of sEVs; therefore, it was necessary to retrospectively analyze the published related papers to help related researchers or researchers wanting to understand this area grasp the hot spots, sort out the development context, and forecast future trends faster and easier. Bibliometrics was derived from documentary statistics in early 20th century and was developed for the quantitative analysis of massive scientific data through mathematical and statistical methods; the interdisciplinary, integration, and systematicness made it a powerful technique for hotspot analysis, cooperative network analysis, evolution path analysis, and visualizing the interesting section within a certain collection of the literature. Up till now, numerous tools, such as VOSviewer [11], Gephi [12], CiteSpace [13], BibExcel [14], and the R package “Bibliometrix” [15], have been exploited to undertake this, and their common fundamental cores were the empirical statistical laws.

Herein, we make use of CiteSpace (version 6.1.R3), VOSviewer (version 1.6.18), and Bibliometrix (R package) to realize a visualization analysis, focusing on the applications of sEVs in the field of bone-related diseases. All three tools could perform a metrological analysis and help provide a comprehensive review of the current status and future prospects in this field.

## 2. Data Sources and Search Strategies

### 2.1. Data Sources

The Web of Science Core Collection (WOSCC) was chosen to provide a reliable database for scientometric analysis. The data sample statistics were collected up to the end of 2023. To improve the precision and representativeness of the data, the search subject was restricted to “Topic”, and for specific search topics, we used synonym, asterisk wildcard, quotation marks, and Boolean operators to achieve a precise search range. The full search terms are available in Appendix A. The type of publications was ‘article’ or ‘review’, without limitations of language/time. A total of 2641 papers in the literature were matched, and the full records and cited references were exported for further analysis.

### 2.2. Search Strategies

We used CiteSpace (version 6.1.R3), VOSviewer (version 1.6.18), and the R package “Bibliometrix” (version 4.3.1) with the online platform (to perform bibliometrics and visualize the data. CiteSpace is a Java (a kind of object-oriented programming language)-based software created by Chaomei Chen in 2004 [16]. The aim was to take advantage of the choice made by experts in the academic field in their papers as a basis for our own identification of the potential of academic literature. CiteSpace provides a variety of metrics of significance, such as citation burstness, betweenness centrality, modularity, and so on. VOSviewer (version 1.6.18.) is another Java-based software, developed in 2009 by VanEck and Waltman [11]. Networks of co-organization, co-citation, co-occurrence, and co-authors can be obtained. The visualization interfaces of VOSviewer are similar to CiteSpace but different forms of co-occurrence, such as time deduction and density maps, are helpful for multidimensional analysis of the selected data. Bibliometrix was created and developed by Massimo Aria and Corrado Cuccurullo [15] in 2017, the packages were built in the R language and it is an open-source tool. Bibliometrix can also perform co-cited analysis, but has unique advances in data reduction and frontier analysis. In this study, CiteSpace was used to generate all of the tables, dual-map overlays for journals, and the keywords time zone chart, and to perform references cluster analysis and bursts analysis. VOSviewer was utilized to conduct most of the co-occurrence analysis and the cluster analyses of institutions and keywords. The Bibliometrix R package was used to export publication numbers and geographical location of countries, generate the annual publication numbers of the top ten authors, and exhibit a three-field plot (Sankey diagram) and trending topics as supplementary materials for visualization analysis.

## 3. Results and Discussion

### 3.1. Quantitative Analysis of Publications

The search results showed that the first article in this field was published in 1997 (Figure 1). Although research on small extracellular vesicles in bone-related diseases started early, for a long time there were not many scientists investigating this area, and only rare research had been published. It was not until 2012 that the number of publications exhibited explosive growth and increased steadily in recent years. Remarkably, the 2013 Nobel Prize in Physiology or Medicine was awarded to James E. Rothman, Randy W. Schekman, and Thomas C. Südhof for their contribution of discovering the regulatory mechanisms of vesicle transport. We considered that this memorable announcement might have contributed to the bursts of studies on sEVs in the last ten years. Moreover, we analyzed the annual publications and found that they fitted well to an exponential curve (Figure 1 red line), which was beneficial for forecasting the potential increase in future academic quantity.

### 3.2. Analysis of Publication Countries

According to CiteSpace 347 countries with 8057 institutions participated in the 2641 articles. We ranked the countries by both publication numbers and by centrality and filtered the top 10 into Table 1. It showed that China had the highest number of publications with a total of 1356, followed by America (451), Italy (162), Germany (98), and Japan (89). Meanwhile, when ranked by centrality, USA displayed the highest degree of centrality (0.4), more than twice that of other countries; right behind were Germany (0.18), China (0.17), Italy (0.11), and India (0.11), which also exhibited high betweenness centrality, and they together greatly promoted the development of this field. The visualization results from VOSviewer help reflect the relationships among countries more directly. We extracted countries with a threshold of five published articles and co-occurred the resulting 49 countries in Figure 2a,b. The nodes’ size was selected to represent the total strength of the link among countries. It was indicated that the USA had the most extensive circle of cooperation, and robust cooperation relationships were established among America, China, and Germany (Figure 2a). In Figure 2b, the color of the nodes represented the date of the first publication in the country, which suggested that Greece, Türkiye, Indonesia, Vietnam, etc., started relevant research relatively late. Bibliometrix was also used for country analysis (Figure 2c,d). In Figure 2c, although China had the largest number of publications, its multiple country publications (MCPs) only occupied a small amount of the total research output, indicating that China lacked intimate academic cooperation with other countries in this field, especially when compared to the USA. The geographical network map in Figure 2d demonstrated that research nations were mostly located in Asia, Europe, Oceania, and North America and that there were still many countries that had not yet dabbled in this novel area.

### 3.3. Analysis of Institutions and Journals

The cooperation relationships between institutions were analyzed using CiteSpace, and the results are shown in Table 2 and Figure 3a. It is worth mentioning that all of the top 10 institutions ranked by publication numbers were universities, suggesting their capabilities of conducting cutting-edge scientific research, while indicating the high threshold in this new frontier. Specifically, Shanghai Jiao Tong University contributed most to the publication numbers, with 121 publications, followed by Central South University (71), Sichuan University (70), and Huazhong University of Science and Technology (60). Regarding the centrality, although Cornell University did not exhibit a large quantity of publications, it exhibited the highest centrality (0.17) of all organizations, followed by Zhejiang University (0.14), the State University System of Florida (0.13), and Harvard University (0.12). The visualization map in Figure 3 could help us analyze the associations among organizations more manifest. The purple circle outside the node in Figure 3a represented high betweenness centrality, which corresponded to the rank order presented in Table 2 (right), and the red circle represented high burstness; this revealed that the National Institutes of Health—USA, the University of Turin, IRCCS Istituto Ortopedico Rizzoli (Scientific Institute for Research, Hospitalization and Healthcare), the University of Texas System, etc., had predominant burst in this field. A clustering analysis of the organizations was carried out using CiteSpace and displayed seven main clusters, as shown in Figure 3c, namely #0 exosome, #1 bone regeneration, #2 regenerative medicine, #3 collagen membrane, #4 bone histology, #5 metastasis, and #6 transcriptomics. Obviously, most of the institutions focused on exosomes, but some paid more attention to certain specific areas and gained great achievements. For example, Harvard University’s main trend was #1, bone regeneration, Cornell University was concerned more about #5, metastasis, and the University of Turin was involved in cluster #3, collagen membrane.

The time deduction visualization map and the density visualization of journals were generated with a minimum publication threshold of 10 documents in VOSviewer and resulted in 51 journals meeting the requirements, of all 724 sources (Figure 3d and Appendix A). Most of the journals belonged to the cell and medicine domain. Stem Cell Research & Therapy exhibited the highest number of citations, at 8095, with the strongest link strength (1415) and the highest number of publications (124), followed by the International Journal of Molecular Science (3825 citations with a link strength of 959) and Stem Cells (3380 citations with a link strength of 435), and these were regarded as the most influential journals in this field. Newly emerging journals are shown in yellow in Figure 3d; it turned out that, recently, there were scientific journals concerned more about biomaterials, such as ACS Applied Materials & Interfaces, Advanced Healthcare Materials, and Bioactive Materials, which all arose after 2021. Based on this, we predicted that the design and development of biomaterials would be a vital hotspot in the application of sEVs to bone-related diseases. We also analyzed the top five journals ranked by publication numbers, and their annual publications between 2012 and 2023 are presented in Figure 3b; all of them reported a steady increase in production, reflecting the popularity of these journals in the field. In particular, a rather high increasing rate could be found for the International Journal of Molecular Science. The dual-map overlay of the journals was generated to reveal the knowledge flow at the journal level (Figure 3e). The main pathways suggested that articles published in the fields of molecular science/biology/immunology were mainly cited by researchers in molecular science/biology/genetics and health/nursing/medicine fields. Likewise, articles in the medicine/medical/clinical fields were mostly cited by articles in the field of molecular science/biology/genetics. On the right of the map, different sizes of ellipses could be seen; a longer vertical axis denoted more publications on the left, and a longer horizontal axis meant more authors, indicating that molecular science/biology/genetics was the most popular field.

### 3.4. Analysis of the Authors

The maps of authors’ collaborative networks were produced by VOSviewer and CiteSpace by retrieving co-authorship information from the publications, and Bibliometrix analyzed the specific authors to further evaluate the activation of the authors in recent years. The statistical data (Appendix A) revealed that Camussi Giovanni, Wang Yan, and Tian Weidong were the top three authors with the most publications, totaling 16, 13, and 12, respectively, and the top three co-cited authors were Thery C., Lai R.C., and Zhang Y. As shown in Figure 4a, the collaborative network of authors exhibited distinct research communities and formed eleven clusters (identified in different colors), and their representative authors were Ravindran Sriram, Zhang Ping, Chopp Michael, Qian Hui, Liu Wei, Zhang Yi, Wang Yan, Li Yi, Tian Weidong, Wang Yang, and Chen Lang, respectively. Through citation analysis (Figure 4b), Peinado Hector was regarded as the most cited author with 4490 citation counts, who had a high H-index of 48 and was constantly dedicated to extracellular vesicles and their metastases in microenvironments. Figure 4c presented the production of the top 10 authors over time. Among the authors, Zhang Y. and Li Y. carried out relevant research earlier than others (their first study began in 2012), and Wang Y. published a total of 56 articles, at a rate of more than 10 articles per year, for the past 4 years. Furthermore, the authors’ productivity coincided well with Lotka’s law (Appendix A), in that the majority of authors (up to 88.6%, Appendix A) only published one or two articles in these fields, which meant most of the researchers were new to this field.

### 3.5. Analysis of References and Keywords

Co-citation analysis is one of the most important functions of CiteSpace. Herein, we performed a cluster analysis of the co-cited references and have displayed the results in Table 3 and Figure 5a. A co-cited relationship referred to two or more references cited by one paper simultaneously, which suggested that there were some common topics between the two and that the cluster of them would help us seek out a main research orientation quickly. As shown in Table 3, the most co-cited reference was published in the Journal of Extracellular Vesicles by Thery C. et al. in 2018, with a local citation number (cited by articles from the database) of 317, which renewed the guideline formulated in 2014 (MISEV, Minimal information for studies of extracellular vesicles) and provided more details on sEV characterization and experimental standards [8]. The second reference was a significant review published in Science by Kulluri R. [17], with a local citation number of 229, in which they focused more on the functions and bio-applications of sEVs in terms of their clinical potential. In particular, they proposed that more animal models should be involved in unravelling the physiological mechanisms of exosomes. The third article was also a review, but restricted sEVs to exosomes derived from MSC (mesenchymal stem/stromal cells); MSCs are one kind of multifunctional cells with easy culture expansion, and now have been used to treat a variety of human diseases [18]. Phinney et al. [19] summarized the evidence that MSC-derived exosomes acted as paracrine mediators in tissue repair and noted that the establishment of the evaluation of the exosomes’ utility and efficiency was crucial in their further clinical trials. All of these highly valuable papers were recommended to researchers in relevant fields to read carefully before conducting the experiments.

With a clustering analysis, the co-cited references could be divided into eight clusters using a log-likelihood ratio (LLR) algorithm (a minimum threshold of 70 articles). These are shown as #0 microparticles, #1 extracellular vesicles, #2 mesenchymal stem cells, #3 cartilage repair, #4 bone metastasis, #5 proinflammatory cytokines, #6 conditioned medium, and #7 diabetic wound healing in Figure 5a. Compared with traditional nanomaterials, sEVs have the advantages of biocompatibility, biodegradability, and low immunogenicity, but the natural sEVs also had the disadvantages of complex components, insufficient targeting efficiency, and functionalization. The engineering modification of sEVs was shown to be of significant importance in broadening the applications of sEVs in bone-related diseases, with numerous references focusing on it. Generally, the engineered sEVs were synthesized using genetic, chemical, or physical methods [27], and the main aim of the modification in bone-related diseases was to construct a drug delivery system to promote the stability and targeting effect of traditional sEVs, to load functional molecules for in vivo tracking, and to load certain drugs to improve therapeutic capacity. Genetic engineering was mainly based on transgene expression, and the objects of genetic operation were the parent cells. The method was rather simple and mature but had an unsatisfactory loading efficiency. The chemical modification was to make use of covalent conjugation reactions to induce chemical groups or functionalized molecules, such as nanoparticles, fluorescent labels, peptides, and antibodies. The click reaction was a perfect biocompatible candidate for it [28]. The chemical modification was aimed at improving efficiency but may have led to unpredictable changes in the sEVs’ properties. Sonication, electroporation, membrane fusion, and freeze–thaw treatments were the common pathways for drug loading in physical modification [29]. The advantage of physical modification was that it greatly guaranteed the integrity of the sEVs, since the method was non-invasive. However, the stability was inferior to that of chemical modification. Suitable engineering alternation strategies could remarkably improve the availability of sEVs. For example, Hu et al. [30] overexpressed CXCR4 in engineered NIH-3T3 cells and efficiently simulated the accumulation of sEVs in bone marrow and promoted osteogenic differentiation. Yan et al. [31] modified the exosomal surface with folic acid (FA)–-polyethylene glycol (PEG)–cholesterol and enhanced the accumulation of drugs at inflammation sites to reduce inflammation in the joints. Due to the limitation of instability and the unclear therapeutic capability of natural sEVs, surface engineering strategies, bone-targeting strategies, and functional yield strategies, etc. were expected for clinical applications of sEVs in bone-targeted therapy (BTP).

The third largest cluster was mesenchymal stem cells (MSCs). MSCs are one of the most promising types of cell therapy due to their multiple linkage differentiation regeneration [32], especially their ability to differentiate into osteocytes and chondrocytes for bone defect repair. MSCs have been reported to play key roles in the promotion of osteogenic differentiation [33], the regulation of osteoclast activity [34], and the capacity for immunoregulation [35] and angiogenesis [36]; thus, MSC-derived sEVs, existing in MSC secretions, were supposed to function as MSCs and contributed to the regeneration of bones [37] without tumorigenicity or immune rejection. MSC-derived sEVs were found to promote the proliferation and migration of chondrocytes in osteoarthritis, inhibit the biological activity of osteoclasts in osteoporosis [38], improve osteogenesis and angiogenesis in bone fractures [39], and enhance axonal growth and regulate inflammatory and immune responses in spinal cord injury repair [40]. Since MSC-derived sEVs were proven to contain numerous growth factors [41] (such as SDF-1, VEGF, and HGF), a diversity of cytokines (such as chemokines and anti-inflammatory cytokines), abundant RNA (such as mRNA [42] (for example, BMP-2), microRNAs [43] (for example, miR-100–5p [44], miR-210 [45]), and LncRNA (for example, LncRNA H19 [46], LncRNA KLF3-AS1 [47])), and proteins (such as HIF-1α [48], CD73 [21]), which all contributed to bone repair, the engineering of MSC-derived sEVs was also popular for specific applications. Ma et al. [49] loaded sEVs with VEGF-A and BMP-2 endogenously through a cellular nano-electroporation system that prominently enhanced angiogenic–osteogenic regeneration. Liu et al. [50] found that hypoxia preconditioning of MSCs could activate *HIF-1α* in MSCs, which further promoted sEV release and enriched *miR-126*, a suppressor of the Ras/ERK pathway. Huang et al. [51] proved that, through the genetic modification of the parental MSCs (the overexpression of *BMP2*), the functional sEVs were shown to have a better regenerative potential than native sEVs in vitro and in vivo. Despite all of these advantages, challenges in clinical applications were formidable. In practical applications, the heterogeneity of sEVs from different sources of tissues and the diversity of isolation and purification methods were needed to be solved to ensure the efficiency of sEVs, which meant a standard procedure was required to be established to guarantee product consistency.

The clusters also revealed that cartilage damage was one of the most researched types of bone disease in this field. Due to the lack of blood supply and innervation, cartilage damage is difficult to repair once formed. Small extracellular vesicles exhibit great superiority in cartilage damage repair because of their abundant content of transcription factors related to bone repair. Zhang et al. [24] claimed that intra-articular injection of exosomes weekly could restore the damaged cartilage and subchondral bone with hyaline cartilage in an osteochondral defect model. When focusing on the possible mechanisms, relevant studies revealed that sEVs played different roles at different levels in cartilage damage. In traumatic cartilage injury, sEVs reduced the pro-inflammatory mediator IL-1β [52], suppressing the NLPR3 pathway [53] and regulating macrophage polarization [54] to reduce inflammation. In ECM resynthesis, sEVs upregulated collagen type II production and downregulated MMP13 protein expression [55], stimulated differentiation of BMSCs into chondrocytes [56], and promoted chondrocyte proliferation [57]. All of these suggested the potential of sEVs to maintain cartilage homeostasis. However, the safety, dosage, frequency, and administration route still need to be considered and further standardized.

References citation bursts meant the references were extensively cited by academics over a certain time period. Herein, visualizations of reference citation bursts were produced using CiteSpace to evaluate the topics of interest over the last ten years. Figure 5b exhibits the top 25 references with the strongest citation bursts ranked by strength, which shows that the article published by Raposo G. and Stoorvogel W. [58] in 2013 possessed the strongest strength and the longest citation period (which began in 2013 and ended in 2018). That review summarized the proposed formation, targeting and function mechanisms, and characteristics of EVs. The second and third strongest articles were both research papers, concluding that MSC-derived exosomes could be a potential adjuvant for the therapy of myocardial infarction but used different sources of mesenchymal stem cells [59,60]. The latest article with the strongest strength was produced by Baglio S.R. et al. [61], focusing on the deep sequencing analysis of the small RNA profile of exosomes from adipose-derived MSCs (ASCs) and bone marrow-derived MSCs (BMSCs). It revealed that tRNA species varied between adipose- and bone marrow-derived MSCs, which marked the focusing on sEVs at the molecular level.

Keywords could be regarded as eyes for catching the theme of the article. Table 4 shows the top 20 keywords with the highest frequency, summarized using CiteSpace. “Extracellular vesicles”, “mesenchymal stem cells”, “exosm”, “stromal cells”, “bone marrow”, “differentiation”, “stem cells”, “expression”, “repair”, and “bone marrow” were the top 10 items among them, showing that the keyword analysis was well in accordance with that produced using Bibliometrix (Appendix A). Moreover, a three-field plot (Sankey diagram) was generated using Bibliometrix (Appendix A) for comprehensive studies among cited references, authors, and keywords. The photography showed that robust links were built between the reference “Raposo G 2013 J Cell Bio” [58] and the author “Wang Y.”. Additionally, references “Kalluri R 2020 Science” [17] and “Thery C 2018 J Extracell Vesicles” [8] contained most of the presented authors, with a flow number of 19. As for relationships between authors and keywords, the author “Li Y.” had the strongest links with the keywords “extracellular vesicles”. Except for “exosomes”, “exosome”, and “extracellular vesicles”, which referred to the same subject, the keywords “angiogenesis”, “mesenchymal stem cells”, “tissue engineering”, and “osteoarthritis” were also broadly studied among these authors.

Keyword bursts from 2013 to 2023 (Figure 6b, left) indicated that “microvesicles” had the highest explosion intensity (15.39) with an early explosion time (began in 2013), since “microvesicles” was first used to refer to membrane vesicles secreted by cells [62], before more detailed classification and isolation techniques appeared. “Human bone marrow” and “in vitro” were keywords with the second highest and third highest burst strengths, indicating that the application of sEVs in bone-related diseases was still at an early stage and highly dependent on the in vitro model. When ranked by timeline (Figure 6b, right), “growth factor”, “animal models”, “endothelial cells”, and “matrix” were the latest words, suggesting that related studies had transformed from in vitro to in vivo and were focused more on cytokines and cell microenvironments for mechanism studies.

We also performed a co-occurrence analysis of keywords using VOSviewer. Figure 6a exhibits a total of 218 keywords, and four main clusters were identified, each representing a unique research orientation in the field. The yellow cluster, which contained keywords such as “mesenchymal stem cells”, “tissue engineering”, “scaffolds”, “bone regeneration”, and “drug delivery”, put a particular emphasis on the application strategies of sEVs in bone-related diseases, especially through tissue engineering techniques. The green cluster, compromising keywords like “differentiation”, “proliferation”, “expression”, “angiogenesis”, and “osteoclast”, was concerned more about the impact and mechanisms of sEVs in cell activities and cell regulations. The red group, namely “stromal cells”, “adipose tissue”, “stem cells”, “endothelial progenitor cells”, “in vitro”, etc., consisting of various types of cells, indicated a concentration on cell therapy and its therapeutic effect in in vitro studies. The last blue-labeled groups involved plenty of injury symptoms, such as “stroke”, “apoptosis”, “oxidative stress”, and “autophagy”, and were focused more on physiological characteristics and physiological mechanisms in bone-related diseases. All four clusters displayed different core concerns of sEVs in the realms of bone-related diseases.

### 3.6. Analysis of Trends and Frontiers

Here, time deduction map analysis and the time sequence of keywords were used to reflect and predict trends and outlooks of the field. The timeline zone map (Figure 7a) produced using CiteSpace reflected the hot spots in individual years, in which the font size represented the frequency of keywords. It revealed that the terms most mentioned in recent years were “senescence”, “osteoarthritis”, “hydrogel”, “scaffolds”, and “platelet-rich plasma”. We also co-occurred keywords using time deduction map in VOSviewer as an auxiliary verification (Figure 6b); keywords in yellow represented new focuses in this field, including the hotspots “senescence”, “biomaterials”, “bone tissue engineering”, “nanoparticles”, “hydrogel”, “osteoarthritis”, “osteogenesis”, and “platelet-rich plasma”, which were well in accordance with the timeline zone map. We further investigated the trending topics through the analysis of author keywords in the last ten years using Bibliometrix (Appendix A), through which we found that the latest terms were “periodontitis”, “pyroptosis”, “tendon–bone healing”, “bone regeneration”, “bone marrow mesenchymal stem cells”, and “osteogenesis”, and the most frequent terms except for synonyms of exosomes were “mesenchymal stem cells”. We then briefly discussed four research frontiers extracted from this software, namely tissue engineering, platelet-rich plasma, osteoarthritis, and senescence.

The traditional drug delivery pathway of sEVs in clinical therapy is intravenous injection or oral administration. Compared with liposomes or polymetric nanoparticles, sEVs have a rather long circulating half-life and the ability to cross the blood–brain barrier [63]. However, as relevant studies developed in depth, sEVs were found to be quickly cleared in blood vessels [64] and encapsulated into parenchymatous organs such as the liver, spleen, and lungs [65,66]. When delivered by oral administration, sEVs were accumulated and absorbed in the intestinal tract [67]. Thus, achieving effective local administration was a critical problem in expanding the application of sEVs, especially for organic diseases. Tissue engineering was an important component of regenerative medicine. Bone tissue engineering (BTE) [68] took advantages of engineering, chemistry, and biology to fabricate bioactive materials and to replace damaged tissues or organs [69]. An ideal artificial bone tissue consisted of three parts: a well-defined scaffold, seed cells (usually stem cells), and cytokines, such as growth factor and bioactive peptide [70]. The integrated manipulation of sEVs to replace traditional seed cells was considered a novel cell-free therapeutic treatment, which greatly avoided the defects of low homing efficiency, low survival rate [71], changes in phenotype, and tumor-forming potential [72]. For the construction of scaffolds, hydrogels with a three-dimensional network played a predominant role because of their excellent biocompatibility and mouldability. There were various types of hydrogels, consisting of different forms of crosslinking. Among the chemical couplings, gelatin methacrylate (GelMA) attracted much attention because of its injectable and UV (ultraviolet radiation)-cross-linked properties. Chen et al. [73] fabricated a 3D-printed cartilage extracellular matrix (ECM)/gelatin methacrylate (GelMA)/exosome scaffold and found that it performed well in osteochondral defect regeneration. Hu et al. [74] combined GelMA hydrogel with laponite nanoclay to enhance the mechanical properties of the scaffolds and successfully achieved the sustained release of sEVs, for the further formation of glycosaminoglycan, an extracellular matrix, and collagen II in cartilage repair. In addition, other natural polymer materials, such as collagen [75], chitosan [76], fibroin [77], hyaluronic acid [78], and synthetic polymer materials, such as polylactic acid–hydroxy acetic acid copolymer [79] (PLGA) and polyethylene glycol (PEG) [80], were also followed with interest for the construction of sEV-loading hydrogels for tissue repair and regeneration [81].

Platelet-rich plasma is a multifunctional platelet concentrate centrifuged from whole blood, in which the platelet concentration is three to five times that of physiological blood. Plentiful growth factors were found to be secreted by activated platelets and exhibited extraordinary performance in promoting cell proliferation and differentiation, facilitating angiogenesis and anti-inflammation properties, enabling it to be widely used in wound healing [82], bone regeneration [83], and the repair of acute muscle, tendon, ligament, nerve, and cartilage injuries, etc. [84]. However, the main limitation in clinical situations was that only autologous platelets were required because of the potential for immunological rejection. In addition to the secretion of growth factors, platelets also secreted small extracellular vesicles, so-called platelet-rich plasma-derived exosomes (PRP-Exos), which contained large numbers of active factors and nucleic acids and displayed similar properties to the source cells [85]. PRP-Exos were proven to have little immunogenicity or tumorigenicity, which made them even suitable for tissue repair and regeneration. In 2014, PRP-Exos were first extracted from platelet lysate by Torreggiani et al. [86] and showed a significant dose-dependent increase in cell proliferation and the migration of BMSCs. Henceforth, the study of PRP-Exos became an emerging trend among researchers, especially in the field of bone-related diseases. Tao et al. [87] applied PRP-Exos to an osteonecrosis of the femoral head (ONFH) model and indicated that PRP-Exos could prevent glucocorticoid-induced apoptosis through the Akt/Bad/Bcl-2 signal pathway under endoplasmic reticulum (ER) stress. Zhang et al. [88] employed thermosensitive gel to deliver PRP-Exos; the PRP-Exo-incorporated gel not only prolonged the release of exosomes, but also induced endogenous BMSCs to the injury site and reduced inflammation in subtalar osteoarthritis (STOA). Although the present research suggested the obvious tissue repair potential of PRP-Exos, the molecular mechanism was still unclear and lacked sufficient reporting evidence. More comprehensive and mechanical investigations were desired to prove its efficiency and safety.

Osteoarthritis (OA) is one of the most common painful bone diseases and affects the quality of life seriously. The typical symptoms of osteoarthritis include arthralgia [89], ankylosis [90], and osteoproliferation [91], and the main pathological feature is the degeneration of the articular cartilage [92]. There is no specific medicine for the curation of osteoarthritis due to its unclear and complicated pathogenesis. Traditional drugs [93], such as non-steroidal anti-inflammatory drugs (NSAIDs) and intraarticular corticosteroid injections, can only help relieve pain and delay disease progression; therefore, it is urgently needed to develop an effective and personalized therapy for OA. As mentioned before, MSCs-derived sEVs have shown great therapeutic potential for cartilage damage repair [94]. It was reported [95] that sEVs isolated from human adipose tissue-derived MSCs (hASCs) exhibited chondroprotective functions by reducing the production of inflammatory mediators (TNF-α, IL-6, PGE2, and NO), enhancing the production of the anti-inflammatory cytokine (IL-10) and collagen II, and decreasing the release of MMP activity and the expression of MMP-13. Zhang et al. [21] extracted sEVs from human embryonic stem cell-derived MSCs [24] and demonstrated that the regeneration efficiency of exosomes was achieved through a coordinated mobilization of multiple cell types and the activation of several cellular pathways. It is well known that the external conditions of cell culture can affect the secretion and characteristics of sEVs. As a common means of handling, hypoxia preconditioning was used to fabricate functional sEVs, and it was concluded that the operation could induce angiopoiesis mediated by ASCs [50] and effectively stimulate the proliferation and osteogenic differentiation of BMSCs [96]. Since miRNAs played an important role in bone generation, making use of genetic engineering or other methods to load different miRNAs into sEVs was another popular strategy for the treatment of OA. In this case, sEVs were considered a natural drug delivery platform, similar to that of liposomes. Tao et al. [23] overexpressed miR-140-5p [97] in synovial mesenchymal stem cells (SMSCs) and found that this modification reduced the side effect of the original exosomes and successfully prevented OA in an early stage of an OA model. Mao et al. [98] isolated exosomes from miR-92a-3p-overexpressing human mesenchymal stem cells and revealed that exosomal miR-92a-3p could act as a Wnt inhibitor and regulate cartilage development. Other strategies, such as embedded sEVs with biomaterials, which has been discussed above, are not described here. Despite the fact that researchers have made great efforts to apply sEVs in the curation of OA, clinical evaluations are rare, and more exploration using severe and chronic OA models should be performed for widely ranging applications.

Senescence was found to be the latest keyword. It has been proven that aged-related bone loss, osteoarthritis progression, intervertebral disc degeneration, etc. are closely associated with cell senescence. As one kind of critical medium in cell communication, sEVs are supposed to have important roles in cellular aging. Thus, one natural idea was to make use of the specificity of sEVs to design potential biomarkers. The senescence of MSCs was supposed to be a contributing factor to aging, with a typical phenotype [99,100] of enlarged morphology, decreased differentiation potential, the reduced expression of surface antigen markers, and the expression of senescence-associated β-galactosidase (SA-β-gal) and senescence-associated secretory phenotype (SASP), and it has been proven that the senescent MSC possessed a specific secretome [101,102]. In one study [103], it was found that senescent late passage (LP) MSCs secreted higher levels of MSC-MVs with smaller sizes than early passage (EP) MSCs, and the LP MSC-MVs had an obvious weaker ability of osteogenesis than the EP MSC-MVs. Combined with miRNA analysis, it was concluded that the integrated characteristics of MSC-MVs could dynamically reflect the senescence state of MSCs. A more recent study [104] carried out transcriptomic and proteomic profiles of different MSCs and found that fetal MSCs had a better capacity for bone formation and regeneration than adult MSCs. Coincidentally, Wijesinghe et al. [105] utilized similar approaches to identify different expression levels of miRNAs, tRNAs, and proteins, and verified that senescent EVs could induce senescence in healthy cells. M. Varela-Eirín et al. [106] promoted considering sEVs positive for Cx43 as a new biomarker of disease progression and as a new target to treat OA, since they found that sEVs containing Cx43 could induce senescence and activate cellular plasticity in target cells. However, the characterization analysis of sEVs itself was complex; thus, at present, more research is focused on expression differences between sEVs and senescent cells rather than selecting one characteristic as a marker for senescence. Another interesting research direction was to make use of sEVs in the curation of age-related disorders. Gong et al. [107] claimed that human embryonic stem cell-derived small extracellular vesicles (hESC-sEVs) rejuvenated senescent BM-MSCs both in vitro and in vivo; the inside proteins activated several classical pathways (Wnt, Sirtuin, PTEN, and AMPK) involved in reversing cell senescence and promoting osteogenic differentiation. Sun et al. evaluated the therapeutic effects of Adipo-sEVs on intervertebral disc degeneration (IVDD) in rats and explored that Adipo-sEVs rejuvenated senescent nucleus pulposus cells and endplate cells by delivering NAMPT and activating NAD+ biosynthesis and the Sirt1 pathway. As a newly arisen research direction, more mechanism studies and relevant in vivo studies need to be developed in the future.

## 4. Conclusions and Prospects

Small extracellular vesicles have been proven to have similar functions to their parent cells without the defects of potential tumorigenicity, which make them a great candidate for regenerative medicine. Due to the complex causes and weak regeneration ability of bone-related diseases, traditional drug therapy was unable to satisfy the need of bone repair. Meanwhile, studies of sEVs in bone diseases have developed rapidly and achieved gratifying results. In this review, we made use of bibliometric analysis to summarize and forecast the development, hotspots, and trends in this field. CiteSpace VOSviewer, and Bibliometrix were employed, and the main findings could be considered to be two parts: the overview of the background and the research progression.

The overview of the background consisted of the publication numbers, countries, institutions, journals, and authors. An exponential curve of publication numbers was identified as beginning in the 1990s, and it was not until 2013 that the publication numbers presented an explosive growth. China, America, and Italy had the most publications, and robust cooperative relationships were established among America, Germany, and China. Numerous institutions participated in this research area and most of them were universities. The top three productive institutions were Shanghai Jiao Tong University, Central South University, and Sichuan University, and seven clusters were found through clustering analysis. The most cited journals were Stem Cell Research & Therapy, Stem Cells, and the International Journal of Molecular Science. Articles published in the fields of molecular science/biology/immunology were mainly cited by researchers in molecular science/biology/genetics and health/nursing/medicine fields. The top three co-cited authors were Thery C., Lai R.C., and Phinney D.G., and the most cited author was Peinado Hector. Authors from China exhibited the characteristic of high yield and Wang Y. had an annual publication number of more than 10.

We conducted a detailed analysis of the hot topics and latest trends in this field of research. The co-occurrence analysis divided the field into four portions: cell sources, application strategies, mechanisms, and corresponding symptoms in bone-related diseases. Clustering analysis showed that the most popular cell sources of sEVs were mesenchymal stem cells, including bone marrow mesenchymal stem cells, adipose mesenchymal stem cells, and umbilical cord mesenchymal stem cells, and the timeline zone map exhibited the trend of studies of senescence, which could be referred to as the newest study interest. To maximize the effectiveness of using sEVs, several strategies were applied under specific conditions. The engineering modification of sEVs was widely studied. In recent years, combining sEVs with biomaterials became a new strategy for bone disease treatment, especially the combination of scaffolds or hydrogels to locate sEVs and realize sustained release. The repair mechanism of sEVs in bone-related diseases were mostly at the cellular level and focused on cell activation and the regulation of relevant factors, the representative keywords for which included “differentiation”, “proliferation”, “expression”, “angiogenesis”, “messenger RNA”, “growth factor”, etc. The in vivo studies began in 2016 and gradually moved towards modelling, wherein mice were the most commonly used animal models. For specific bone-related areas, “bone histology”, “metastasis”, and “transcriptomics” were studied by most of the institutions; keywords like “spinal cord injury”, “stroke”, “osteoporosis”, and “fibrosis” also put an emphasis on different types of bone diseases, but the most concerning one was “cartilage repair”, found using co-cited references clustering analysis. The latest trend of applications in bone-related diseases was in osteoarthritis, in which sEVs usually played the role of a drug delivery platform.

The formation and regulation mechanisms of bone-related diseases are complicated, and most of the current research still remains at the cellular level. For further studies, new insights into mechanisms among organ systems under physiological and pathological conditions are needed to establish the network of interactions between various factors. On the other hand, although the discovery of sEVs is encouraging, it is a great challenge to guarantee the safety, the homogeneity, and the efficiency of using sEVs as a kind of regenerative medicine. A standard technological process containing a clear cell source, a normative isolation method, and a stable and reproducible production are urgently needed. Based on the existing articles, we recommend MSCs and PRP as optimized cell sources for EV production. To overcome the natural instability of sEVs, EV engineering strategies and tissue engineering with biomaterials are accessible for increasing the effectiveness of EV therapy. However, in practical clinical trials, the EV pharmacokinetics need to be considered, especially for modified ones; the relatively poor mechanical performance and the operability of hydrogels in practical use need to be improved, and larger animal models should be studied before they are used in clinical practice. There is still a long way to go for widespread adoption, but we believe that with the joint efforts of scientists in the fields of biology, medicine, bioengineering, chemistry, and material science, using functional sEVs for the curation of bone-related disease will be realized in the near future.

## 5. Limitations

Several limitations should be noted in our data screening. First, due to the variation of appellation of small extracellular vesicles, we included as many different expressions referred to small extracellular vesicles as possible; thus, the search results could not exactly exclude other subtypes of extracellular vesicles. This was a conflict between searching accurately and searching comprehensively. Second, since our database was only from WoSCC, papers published in other periodical databases relevant to this field may have been missed. Third, restricted to the statistical method of the software, we focused more on articles with high citation frequencies, but the newly published high-quality articles were not discussed, which could have given us a supplementary perspective of the research frontier in this fresh research area. Despite these defects, we believe that this study is helpful in summarizing the general trends and directions in this field.

## Figures and Tables

**Figure 1 cells-13-00904-f001:**
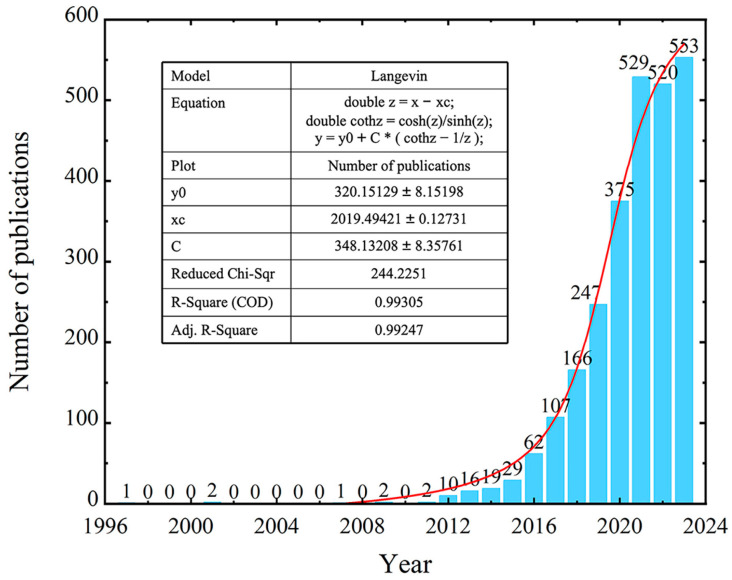
Annual publication number from 1997 to 2023 at WOSSC (inset: nonlinear fitting of publication numbers during 1997–2023).

**Figure 2 cells-13-00904-f002:**
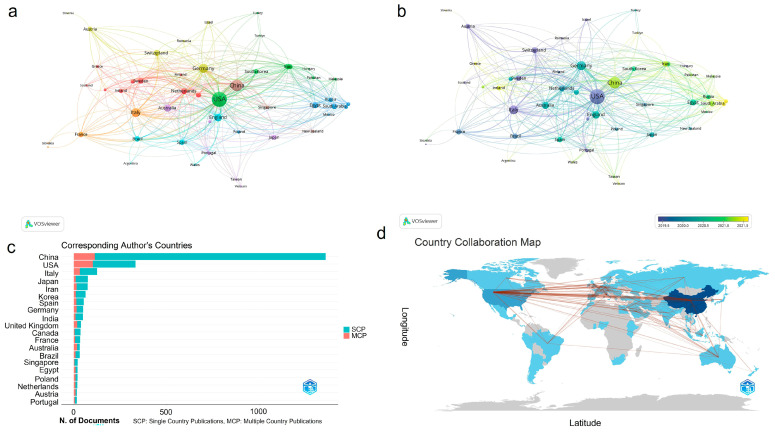
Analysis of national cooperative relationships. (**a**) The co-occurrence network of countries conducted by VOSviewer with a threshold of five documents; (**b**) corresponding time deduction map of (**a**); (**c**) top 20 countries with the largest number of documents, with analysis conducted by Bibliometrix; (**d**) country collaboration map produced using Bibliometrix.

**Figure 3 cells-13-00904-f003:**
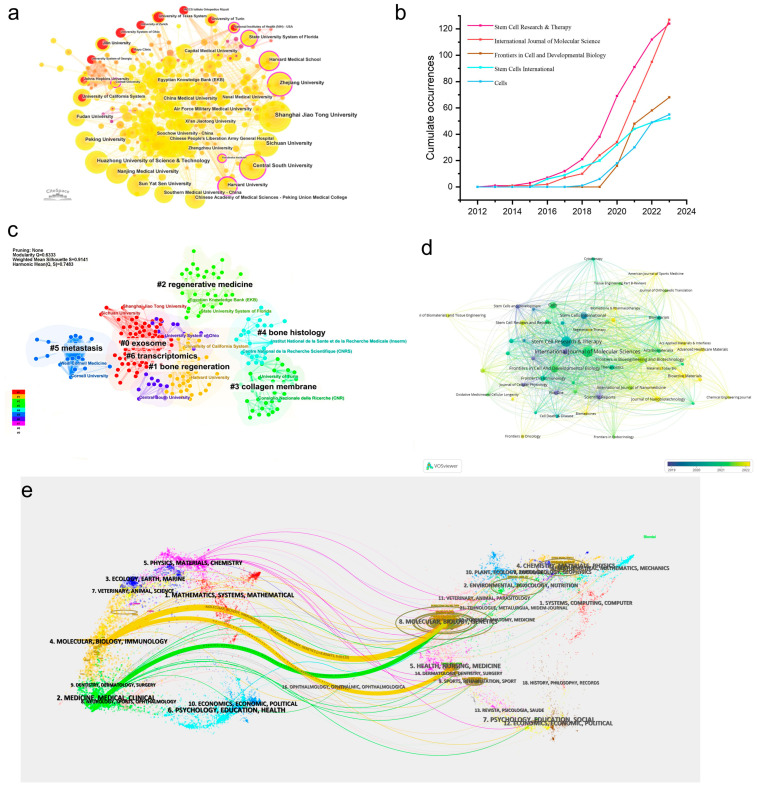
Analysis of institutions and journals. (**a**) Network map of organizations based on CiteSpace; (**b**) annual publications of the top five journals between 2012 and 2023, reproduced from Bibliometrix; (**c**) cluster analysis performed using CiteSpace; (**d**) visualization map of citing journals produced using VOSviewer with a threshold of 10 documents; and (**e**) dual-map overlay of journals made using CiteSpace. The nodes represent the references, the labels represent the discipline, and the line represents the citation pathway. The left is the citing journals and the right indicates the co-cited journals.

**Figure 4 cells-13-00904-f004:**
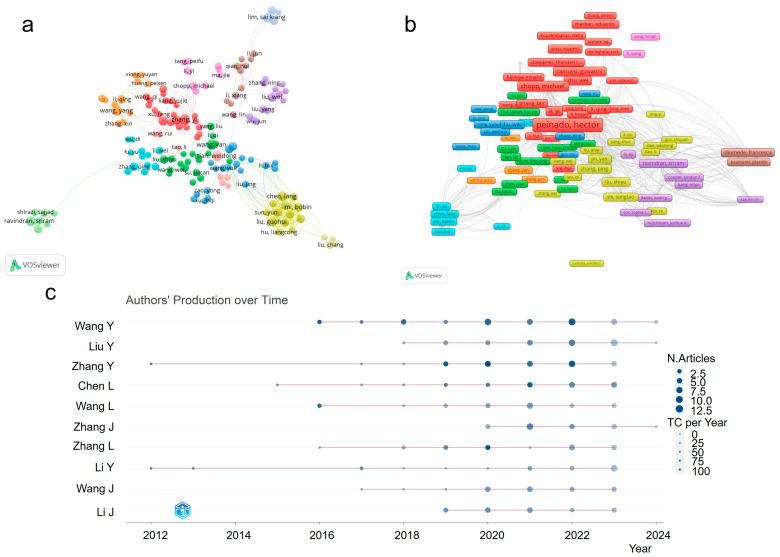
Analysis of authors. (**a**) Collaborative network of authors conducted using VOSviewer with a minimum threshold of five documents of an author through co-authorship analysis; (**b**) visualization map of authors, similar to (**a**), but made using citation analysis, wherein the size of the frames represented the citation counts; (**c**) authors’ production over time using Bibliometrix. The shade of the color reflected the citation counts, and the size of the circle represented the number of publications.

**Figure 5 cells-13-00904-f005:**
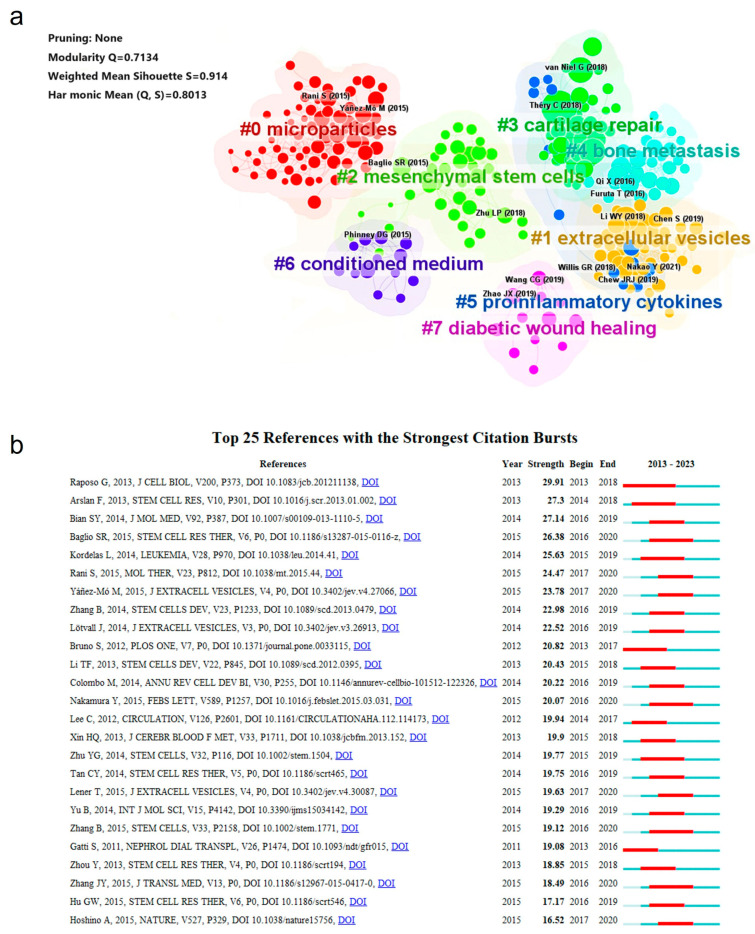
Analysis of references conducted by CiteSpace. (**a**) Clustering of co-cited references; (**b**) top 25 references with the strongest citation bursts. Redline represented the burst period.

**Figure 6 cells-13-00904-f006:**
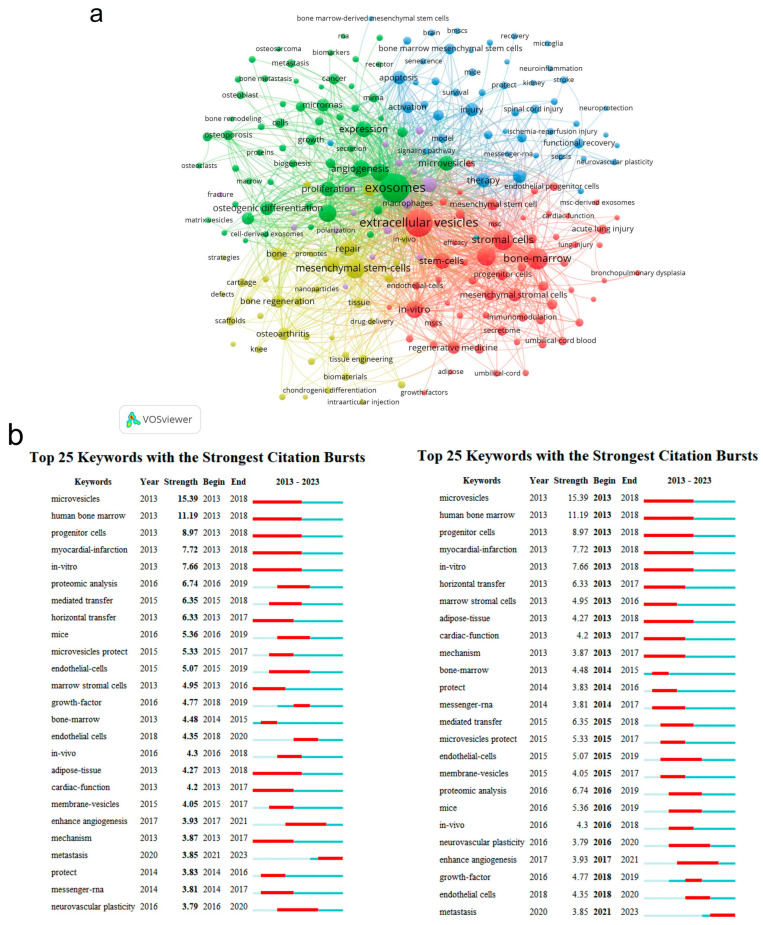
Keyword analysis. (**a**) Co-occurrence map of keywords conducted using VOSviewer with a threshold of 20; (**b**) top 25 keywords with the strongest citation bursts based on CiteSpace, through different kinds of sorting. Right: by strength; left: by start time. Lines in red meant the burst period.

**Figure 7 cells-13-00904-f007:**
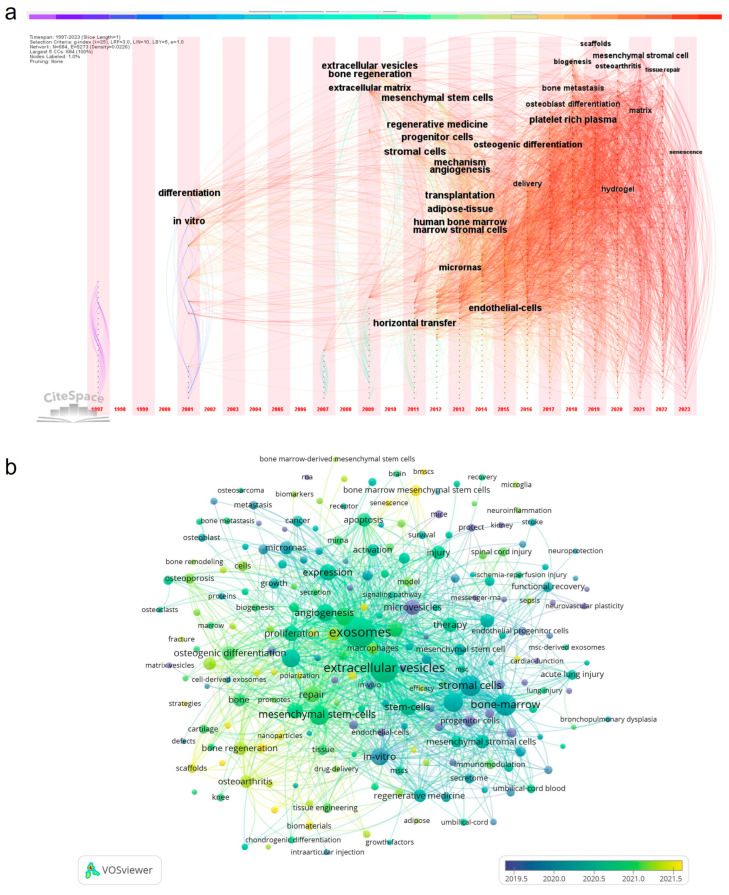
Analysis of trend topics and frontiers. (**a**) Time zone chart of keywords analyzed using CiteSpace; (**b**) time deduction map of keywords analyzed using VOSviewer.

**Table 1 cells-13-00904-t001:** Top 10 countries ranked by publication numbers and by centrality.

Rank	Countries Ranked by Publication Numbers				Countries Ranked by Centrality			
	Counts	Centrality	Year	Country	Counts	Centrality	Year	Country
1	1356	0.17	2013	China	451	0.4	1997	USA
2	451	0.4	1997	USA	98	0.18	2012	Germany
3	162	0.11	2012	Italy	1356	0.17	2013	China
4	98	0.18	2012	Germany	162	0.11	2012	Italy
5	89	0.07	2013	Japan	62	0.11	2014	India
6	81	0.07	2016	Iran	66	0.1	2015	England
7	76	0.03	2013	South Korea	66	0.09	2012	Spain
8	66	0.1	2015	England	35	0.08	2015	Egypt
9	66	0.09	2012	Spain	89	0.07	2013	Japan
10	62	0.11	2014	India	81	0.07	2016	Iran

**Table 2 cells-13-00904-t002:** Top 10 institutions ranked by publication numbers and by centrality.

Rank	Institutions Ranked by Publication Numbers				Institutions Ranked by Centrality			
	Counts	Centrality	Year	Institutions	Counts	Centrality	Year	Institutions
1	121	0.06	2016	Shanghai Jiao Tong University	8	0.17	2011	Cornell University
2	71	0.1	2014	Central South University	60	0.14	2017	Zhejiang University
3	70	0.04	2019	Sichuan University	29	0.13	2017	State University System of Florida
4	60	0.03	2015	Huazhong University of Science and Technology	39	0.12	2014	Harvard University
5	60	0.14	2017	Zhejiang University	13	0.12	2009	National Institutes of Health (NIH)—USA
6	56	0.02	2018	Sun Yat Sen University	6	0.12	2020	Karolinska Institutet
7	52	0.01	2018	Nanjing Medical University	71	0.1	2014	Central South University
8	48	0.03	2014	Fudan University	47	0.1	2020	Southern Medical University—China
9	48	0.04	2018	Peking University	34	0.1	2015	Egyptian Knowledge Bank (EKB)
10	47	0.1	2020	Southern Medical University—China	22	0.09	2007	University System of Ohio

**Table 3 cells-13-00904-t003:** Top 10 co-cited references analyzed using CiteSpace.

Citing Counts	Years	Information about the Reference
317	2018	Théry C, **2018**, *J. Extracell. Vesicles*, [8] DOI 10.1080/20013078.2018.1535750
229	2020	Kalluri R, **2020**, *Science*, [17] DOI 10.1126/science.aau6977
179	2017	Phinney DG, **2017**, *Stem Cells*, [19] DOI 10.1002/stem.2575
177	2018	Van Niel G, **2018**, *Nat. Rev. Mol. Cell Bio.*, [20] DOI 10.1038/nrm.2017.125
170	2018	Zhang SP, **2018**, *Biomaterials*, [21] DOI 10.1016/j.biomaterials.2017.11.028
145	2018	Li WY, **2018**, *Acs Appl. Mater. Inter.*, [22] DOI 10.1021/acsami.7b17620
119	2017	Tao SC, **2017**, *Theranostics*, [23] DOI 10.7150/thno.17133
111	2016	Zhang S, **2016**, *Osteoarthr. Cartilage*, [24] DOI 10.1016/j.joca.2016.06.022
108	2019	Pegtel DM, **2019**, *Annu. Rev. Biochem.*, [25] DOI 10.1146/annurev-biochem-013118-111902
107	2016	Qi X, **2016**, *Int. J. Biol. Sci.*, [26] DOI 10.7150/ijbs.14809

**Table 4 cells-13-00904-t004:** Top 20 keywords, analyzed using CiteSpace.

Rank	Counts	Centrality	Year	Keywords
1	1063	0.3	2014	extracellular vesicles
2	772	0.23	2012	mesenchymal stem cells
3	545	0.11	2014	exosm
4	499	0.12	2012	stromal cells
5	348	0.12	2012	bone marrow
6	330	0.04	2012	differentiation
7	311	0.1	2012	stem cells
8	284	0.07	2012	expression
9	253	0.06	2017	repair
10	240	0.04	2012	bone marrow
11	220	0.16	2009	in vitro
12	212	0.06	2012	proliferation
13	209	0.05	2012	regeneration
14	208	0.04	2013	angiogenesis
15	189	0.03	2012	therapy
16	188	0.04	2014	osteogenic differentiation
17	186	0.07	2012	microvesicles
18	173	0.03	2015	mesenchymal stromal cells
19	164	0.01	2012	transplantation
20	162	0.03	2013	injury

## Data Availability

The raw data can be directly obtained from the Web of Science Core Collection (WoSCC) database.

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
