# Peer review of "Visualization Analysis of Small Extracellular Vesicles in the Application of Bone-Related Diseases"

_cells, 2024, doi:10.3390/cells13110904_

Round 1

Reviewer 1 Report

Comments and Suggestions for Authors

I have to admit that this paper caught me off guard. I think we, as scientists, need to analyze all the parameters leading to a phenomenon like EVs but in my opinion, we should try to bring some concrete facts on the table of science. This is a kind of review not really interesting to me. Not bad but also not good. What is the aim of this paper at the end of the day?

Author Response

We feel so sorry that the manuscript didn’t catch your eyes. Traditional reviews surely contain concrete discussions by the output of expert opinions, but we could know only a tiny part of the research area through reading these, which are not very friendly to new researchers or people want to know the general status of the area. This manuscript takes mathematical tools to do scientific analysis. On the one hand, it avoids human bias as much as possible. On the other hand, we could gain more information, such as finding the authors you concern about, their annual publications, their network with other authors, who is active in recent years, is there any institute dominant the area? Which paper are recommended to read? What topics are hotspots? We also discussed specific points that are indicated vital in the area, especially in the trend and frontiers part.

PS: the date doesn’t mean anything, it’s just because we collected the data that day as it takes a period of time to analyze them. In the revised manuscript, we renewed the data to the end of 2023 and have made extensive modifications to our manuscript and supplemented extra data to make our results convincing. We hope the revised manuscript could be acceptable for you.

Reviewer 2 Report

Comments and Suggestions for Authors

In this review, the authors employed scientometric synthesis method to conduct a retrospective analysis of sEVs in the field of bone-related diseases. This approach offers new perspectives, enabling researchers to understand the historical development of this area and identify novel research directions. The review presents a fresh way to comprehend this field, which is commendable. However, there are sections that raise concerns and might benefit from further analysis and discussion. Thus, I recommend a major revision with the following suggestions:

1. Abstract, please include specific conclusions drawn from the retrospective analysis directly in the abstract to provide a clear summary of the findings.

2. Introduction: given that the MISEV 2023 guidelines have been released recently, an update reflecting these new standards would be beneficial in the introductory section.

3. Keywords and Analysis: in consideration of the field of bone-related diseases, it would be beneficial to include "senescence" as a keyword and integrate it into the analysis discussions.

3. As we are nearing mid-2024, I recommend including all relevant data up to the end of 2023 to ensure the review's comprehensiveness.

4. Can the authors expand the discussion on sEVs derived from MSCs for bone regeneration? Specifically, it would be valuable to include analyses of bioactive components, particularly engineered EVs that carry cargoes such as proteins (doi.org/10.1016/j.actbio.2020.04.017), miRNA (doi.org/10.1016/j.actbio.2022.07.015)and mRNA (doi.org/10.1002/advs.202302622). These are just examples and please include more related reference.

5. The authors have analyzed sEVs in the context of bone-related diseases, focusing on preclinical research. An area for further exploration could be the connection between this preclinical work and current clinical trials or commercial activities. For instance, it appears that China has a significant number of preclinical cases. Does this imply that China is also leading in the number of clinical trials and the development of commercial products related to this research? This relationship merits closer examination to understand the translational impact of these preclinical studies.

Comments on the Quality of English Language

Minor editing of English language required

Author Response

Thank you very much for your comments concerning our manuscript entitled "Visualization analysis of small extracellular vesicles in the application of bone-related diseases"(ID: cells-2983502). Those comments are all valuable and very helpful for revising and improving our paper, as well as the important guiding significance to our researches. We have studied comments carefully and have made correction which we hope meet with approval, and our answers were marked in red color.

  1. Abstract, please include specific conclusions drawn from the retrospective analysis directly in the abstract to provide a clear summary of the findings.

We are very sorry for our negligence of the conclusions in abstract. We have made correction according to the Reviewer’s comments.

  1. Introduction: given that the MISEV 2023 guidelines have been released recently, an update reflecting these new standards would be beneficial in the introductory section.

Thanks for the advice, we uploaded the guidelines in our manuscript. The MISEV 2023 standardized different appellations of particles from cells and recommended specific separation methods according to their advantages and didsadvantages.

  1. Keywords and Analysis: in consideration of the field of bone-related diseases, it would be beneficial to include "senescence" as a keyword and integrate it into the analysis discussions.

We agreed that “senescence” played important roles in bone-related diseases, we re-analyzed the data part and found the word in the newly emerged keywords. Thus, we added the discussion of senescence in the “Analysis of Trends and Frontiers”.

  1. As we are nearing mid-2024, I recommend including all relevant data up to the end of 2023 to ensure the review's comprehensiveness.

According to the reviewers’ comments, we have renewed all of the data to the end of 2023 in the manuscript and supplementary information and have made extensive modifications to make our results convincing.

  1. Can the authors expand the discussion on sEVs derived from MSCs for bone regeneration? Specifically, it would be valuable to include analyses of bioactive components, particularly engineered EVs that carry cargoes such as proteins (doi.org/10.1016/j.actbio.2020.04.017), miRNA (doi.org/10.1016/j.actbio.2022.07.015)and mRNA (doi.org/10.1002/advs.202302622). These are just examples and please include more related reference.

We sincerely appreciate the valuable comments. We have expanded the discussions of components in sEVs in the discussion part of MSC-derived sEVs, and have carefully added more references to reinforce the engineering of MSC-derived sEVs in our revised manuscript.

  1. The authors have analyzed sEVs in the context of bone-related diseases, focusing on preclinical research. An area for further exploration could be the connection between this preclinical work and current clinical trials or commercial activities. For instance, it appears that China has a significant number of preclinical cases. Does this imply that China is also leading in the number of clinical trials and the development of commercial products related to this research? This relationship merits closer examination to understand the translational impact of these preclinical studies.

According to our data analysis, China did possess a large amount of researchers and papers in the field of bone-related diseases. We totally agreed that more valuable contents should be involved in exploring the connection between preclinical work and current clinical trials or commercial activities. However, the analysis is currently limited by two aspects: first is the source of data, most of the commercial activities can not be find in a convincible website, and we cannot guarantee the authenticity and accuracy of online information, let alone those that were not publicly disclosed. Second is the lack of professional tools. At present, the software we take use of in the manuscript can only analyze literature data in specific formats, but the translational ones may have different forms, such as project application, clinical trial report, etc. we hope in the future, more tools can be employed for the evaluation of technology achievement transformation, but the study is beyond the scope of this report which focuses on literature analysis regarding the applications of sEVs in bone-related diseases.

Round 2

Reviewer 2 Report

Comments and Suggestions for Authors

The authors have satisfactorily addressed my concerns.